# Semi-Automatic MRI Feature Assessment in Small- and Medium-Volume Benign Prostatic Hyperplasia after Prostatic Artery Embolization

**DOI:** 10.3390/diagnostics12030585

**Published:** 2022-02-25

**Authors:** Vanessa F. Schmidt, Mirjam Schirren, Maurice M. Heimer, Philipp M. Kazmierczak, Clemens C. Cyran, Moritz Wildgruber, Max Seidensticker, Jens Ricke, Olga Solyanik

**Affiliations:** Department of Radiology, University Hospital, LMU Munich, 81377 München, Germany; mirjam.schirren@med.uni-muenchen.de (M.S.); maurice.heimer@med.uni-muenchen.de (M.M.H.); philipp.kazmierczak@med.uni-muenchen.de (P.M.K.); clemens.cyran@med.uni-muenchen.de (C.C.C.); moritz.wildgruber@med.uni-muenchen.de (M.W.); max.seidensticker@med.uni-muenchen.de (M.S.); jens.ricke@med.uni-muenchen.de (J.R.); olga.solyanik@med.uni-muenchen.de (O.S.)

**Keywords:** prostatic artery embolization, benign prostatic hyperplasia, PROEMBO trial, MRI of prostate, diffusion MRI, IPSS, ICIQ

## Abstract

(1) Background: To assess the treatment response of benign prostatic syndrome (BPS) following prostatic artery embolization (PAE) using a semi-automatic software analysis of magnetic resonance imaging (MRI) features and clinical indexes. (2) Methods: Prospective, monocenter study of MRI and clinical data of *n* = 27 patients with symptomatic BPS before and (1, 6, 12 months) after PAE. MRI analysis was performed using a dedicated semi-automatic software for segmentation of the central and the total gland (CG, TG), respectively; signal intensities (SIs) of T1-weighted (T1w), T2-weighted (T2w), and diffusion-weighted images (DWI), as well as intravesical prostatic protrusion (IPP) and prostatic volumes (CGV, TGV), were evaluated at each time point. The semi-automatic assessed TGV was compared to conventional TGV by an ellipse formula. International prostate symptom score (IPSS) and international consultation on incontinence questionnaire–urinary incontinence short form (ICIQ-UI SF) questionnaires were used as clinical indexes. Statistical testing in the form of ANOVA, pairwise comparisons using Bonferroni correction, and multiple linear correlations, were conducted using SPSS. (3) Results: TGV was significantly reduced one, six, and 12 months after PAE as assessed by the semi-automatic approach and conventional ellipse formula (*p* = 0.005; *p* = 0.025). CGV significantly decreased after one month (*p* = 0.038), but showed no significant differences six and 12 months after PAE (*p* = 0.191; *p* = 0.283). IPP at baseline was demonstrated by 25/27 patients (92.6%) with a significant decrease one, six, and 12 months after treatment (*p* = 0.028; *p* = 0.010; *p* = 0.008). Significant improvement in IPSS and ICIQ-UI SF (*p* = 0.002; *p* = 0.016) after one month correlated moderately with TGV reduction (*p* = 0.031; *p* = 0.05, correlation coefficients 0.52; 0.69). Apparent diffusion coefficient (ADC) values of CG significantly decreased one month after embolization (*p* < 0.001), while there were no significant differences in T1w and T2w SIs before and after treatment at each time point. (4) Conclusions: The semi-automatic approach is appropriate for the assessment of volumetric and morphological changes in prostate MRI following PAE, able to identify significantly different ADC values post-treatment without the need for manual identification of infarct areas. Semi-automatic measured TGV reduction is significant and comparable to the TGV calculated by the conventional ellipse formula, confirming the clinical response after PAE.

## 1. Introduction

Benign prostatic hyperplasia (BPH) describes the histological diagnosis of benign prostatic enlargement (BPE), a common disease among older men with increasing prevalence from 50% of 50-year-old males to >80% in those over 70. Prostatic enlargement is accompanied by urethral compression with clinical manifestations in the form of lower urinary tract symptoms (LUTS), including weak urinary stream, increased urinary frequency, and nocturia. In addition, bladder outlet obstruction (BOO) due to BPH can result in incontinence, urinary retention, urinary tract infection, hematuria, bladder calculi, and obstructive uropathy. Consequently, these patients not only suffer complications but also a significant decline in quality of life [1,2,3,4,5].

Various treatment options for BPH with distinct safety and efficacy profiles have emerged in experimental and clinical use, including medication therapies, thermotherapy-like transurethral needle ablation (TUNA) or transurethral microwave therapy (TUMT), laser treatments like photoselective vaporization (PVP) or holmium laser enucleation of the prostate (HoLEP), prostatic urethral lift (PUL), transurethral resection of the prostate (TURP), and prostatectomy [4,6,7]. Among the minimally invasive treatments, prostatic artery embolization (PAE) has been established as a low-risk procedure providing high rates of technical success as well as improving qualitative and quantitative measures of LUTS [8,9,10,11,12,13,14]. Magnetic resonance imaging (MRI) presents high-resolution details of the prostate gland [15,16], showing volumetric changes and characteristics of infarction following PAE, as reported by Zhang et al. [17] and Ali et al. [18]. In both of the aforementioned studies, MRI data were reviewed as a means of manual assessment of morphological and volumetric changes. Therefore, in contrast, the purpose of this prospective monocentric study following PAE for the treatment of BPH was to evaluate the morphological and volumetric changes in prostate MRI using dedicated semi-automatic software while defining the potential benefits, as well as to correlate corresponding clinical indexes.

## 2. Materials and Methods

### 2.1. Patients and Study Protocol

This study is a subgroup analysis within the ongoing prospective clinical trial PROEMBO registered with the German Clinical Trials Register (DRKS) (identification number: DRKS00006308) to evaluate the clinical safety and feasibility of Dyna-computed tomography (CT)-guided arterial prostate embolization (d-PAE) in patients with symptomatic BPH. All patients were treated with PAE at our center and all procedures were performed in accordance with the ethical standards of the Declaration of Helsinki. The study was approved by the local ethics committee (Protocol No.: 18-220) and written informed consent was obtained from each patient. Thirty-five patients underwent PAE from July 2018 to December 2019 and met the following inclusion criteria: age ≥ 40, severe BPH-related LUTS, refractory to/contraindicated for medical treatment, international prostate symptom score (IPSS) ≥ 8 (moderate to severe), and unsuitable for surgical treatment. Exclusion criteria were histologically proven prostate/bladder cancer, neurogenic/hypotonic bladder, prior treatment for urinary incontinence, stones, urethral strictures, prostatitis, indwelling catheterization, and acute urinary retention. Patient selection was performed in a multidisciplinary approach by urologists and radiologists. In the present subgroup analysis, 22.9% (8/35) of patients were excluded due to incomplete MRI follow-up. Consequently, 77.1% (27/35) of patients with MRI scans completed at the baseline plus one-, six-, and 12-month follow-ups were analyzed. Two standardized questionnaires, the IPSS and the international consultation on incontinence questionnaire–urinary incontinence short form (ICIQ-UI SF), were assessed at each MR imaging time point.

### 2.2. Prostatic Artery Embolization

Minimally invasive PAE was performed by experienced interventional radiologists (JR, MS, MW) under local anesthesia and conscious sedation. Prior to the procedure, a pelvic CT–angiogram was performed to evaluate the anatomy of the iliac and prostatic arteries. Subsequently, embolization of both sides was performed using 250–355 µm Contour™ polyvinyl alcohol (PVA) embolization particles (Boston Scientific GmbH, Marlborough, MA, USA). Interventional success was defined by arterial stasis. A vascular occlusion system was used in patients with severe voiding symptoms to reduce immobilization time to 4–6 h. Patients were discharged if there were no complications after a routine inpatient stay of 2–3 days according to the study protocol. As part of the PROEMBO trial, overall results [IPSS, ICIQ-UI SF, international index of erectile function (IEFF-5), peak urinary flow rate (Qmax), post-void residual urine volume (PVR), and prostate-specific antigen (PSA) measurements] were assessed at the baseline and at one-, six-, and 12-month follow-ups after PAE. In the present subgroup analysis, we included IPSS and ICIQ-UI SF as clinical indexes due to complete collection of these data for all patients at each follow-up.

### 2.3. MR Imaging Sequences

The MRI sequences and parameters are shown in Table 1.

The prostate MRI was performed using a 3.0 Tesla MRI scanner MAGNETOM Skyra (Siemens Healthineers, Erlangen, Germany). All patients underwent examination in the supine position using a surface phased array coil. The MRI sequences covering the whole prostate and seminal vesicles included axial and sagittal T2-weighted (T2w) turbo spin echo (SE) sequences and axial T1-weighted (T1w) volumetric interpolated breath-hold examination (VIBE) sequences with the Dixon method (in-phase, opposed-phase, fat only, water only). In addition, diffusion-weighted imaging (DWI) using echo planar imaging (EPI) SE sequences and dynamic contrast-enhanced (DCE) images of the prostate were performed.

### 2.4. Semi-Automatic MR Image Analysis

MR image analysis was conducted using the dedicated semi-automatic software tool mint Lesion™ (Mint Medical GmbH, Heidelberg, Germany) to assess changes in signal intensities (SIs) on T1w, T2w, and apparent diffusion coefficient (ADC)-Map images by segmentation of both the central and total prostate gland (CG, TG), respectively (Figure 1). 

Using this software tool, previous segmentations of the CG and TG in T2w images were automatically applied to the corresponding slice levels of T1w axial sequence and ADC-Map. Furthermore, we analyzed volumetric changes (TGV, CGV) and intravesical prostatic protrusion (IPP) using semi-automatic software analysis. Volumetry of TGV was additionally performed by the ellipse formula previously described by Sosna et al. (transverse diameter × craniocaudal diameter × anteroposterior diameter × π/6 [19]. The IPP was assessed at sagittal T2w sequences. All MRI examinations were evaluated by the consensus of one abdominal radiologist (OS) with eight years of experience in prostate MRI and one radiology resident (VFS) with three years of experience.

### 2.5. Statistical Analysis

The statistical analysis was performed using SPSS for Macintosh (IBM IPSS Statistics, Version 25.0, IBM Corp., Armonk, NY, USA). Pre- and post-treatment changes in the SIs, IPP, TGV, and CGV as well as in the clinical indexes (IPSS and ICIQ-UI SF) were examined by ANOVA, including pairwise comparisons using Bonferroni correction and multiple linear correlation analyses. In addition, a subanalysis by baseline TGV ≤ 60 mL and >60 mL was performed using *t*-tests. Normally distributed data were expressed in means ± standard deviation (SD).

## 3. Results

### 3.1. Study Cohort

Male patients presented with a median age of 67 years (range 43–84). At the baseline, the median total PSA was 4.6 ng/dL (range: 0.4–26.1 ng/dL). One patient demonstrated an IPSS of 5, thus not formally meeting the inclusion criteria of IPSS ≥ 8, though he was still included due to a strong wish for treatment. Detailed baseline characteristics of the study cohort are summarized in Table 2.

Interventional success was achieved in all patients (27/27, 100%), including 25/27 (92.6%) patients with bilateral PAE and 2/27 (17.4%) patients with unilateral embolization. One patient (1/27, 3.7%) received two treatment sessions of PAE due to failure of the first procedure. No major adverse events were noted. Minor complications (e.g., transient hematuria, hematospermia, and/or a small amount of rectal bleeding) were self-limiting and disappeared during the first week after PAE.

### 3.2. Volumetric Analyses

Semi-automatic segmentation revealed a mean TGV and CGV of 80.4 ± 10.1 mL and 58.1 ± 9.2 mL at the baseline, respectively. The results of multiple comparisons showed a significant reduction in the TGV between the baseline and one, six, or 12 months after embolization using both the ellipse formula and the semi-automatic method for the total cohort as well as for the subgroups with baseline TGV ≤ 60 mL and >60 mL (Table 3).

There was no significant difference (*p* > 0.05) between the results of both volumetric measurement methods.

After embolization, no significant difference was observed in terms of semi-automatic assessed TGV between one and six months (69.9 ± 8.5 vs. 69.6 ± 8.9 mL, respectively; *p* = 1.00) and one and 12 months (69.9 ± 8.5 vs. 70.2 ± 8.8 mL, respectively; *p* = 1.00). The mean CGV decreased significantly between the baseline and one month after embolization for the total cohort as well as for the subgroups with baseline TGV ≤ 60 mL and >60 mL (Table 4).

There was no significant difference in the CGV between the baseline and six or 12 months after embolization (*p* > 0.05, Table 4).

The median decrease in TGV and CGV for the total cohort was 13.1% (CI 12.1–23.2) and 18.9% (CI 18.2–27.9), respectively, at 12 months.

All patients with demonstrated IPP at the baseline (25/27, 92.6%) experienced a volume reduction of IPP after treatment. The IPP significantly decreased from 19.4 ± 2.3 mm before embolization to post-treatment values of 17.1 ± 2.1 mm (*p* = 0.028), 16.9 ± 1.9 mm (*p* = 0.010), and 17.0 ± 2.1 mm (*p* = 0.008) after one, six, and 12 months for the total cohort (Table 4). In the subgroup with baseline TGV ≤ 60 mL presenting IPP of 14.4 ± 3.2 mm before embolization, there were no significant differences between the baseline and any time point (*p* < 0.005, Table 4).

### 3.3. Changes in Signal Intensities

No patient demonstrated findings of infection/abscess or changes in periprostatic fat. Infarct areas were observed in all patients after embolization and occurred exclusively in the CG (Figure 2).

All patients with bilateral PAE developed infarcts bilaterally (25/27, 92.6%). The SIs were measured on the different MRI sequences using the semi-automatic approach (see Figure 1). Results of multiple comparisons showed significant differences in ADC values (presented in 10^−3^ mm^2^/s) of the CG between the baseline and one month after embolization for the total cohort (1.20 ± 0.26 vs. 1.13 ± 0.23; *p* < 0.001) as well as for the subgroups with baseline TGV ≤ 60 mL (1.16 ± 0.39 vs.; 1.08 ± 0.29; *p* < 0.001) and >60 mL (1.25 ± 0.29 vs. 1.17 ± 0.30, *p* < 0.001). However, for the total cohort and the subgroups, there were no significant differences between the baseline and six or 12 months post-embolization (*p* > 0.05). In terms of the ADC values of the TG, there were no significant differences between the baseline and any time point. Complete data are presented in Table 5.

On T1w images, infarcts were initially hyperintense; correspondingly, the SIs on T1w were most obvious one or six months after embolization. However, statistically, there was no significant difference in T1w SIs of the CG between the baseline and any time point after embolization (*p* = 1.00). Twenty patients (20/27, 74.1%) had a qualitative decrease in T2w SIs relative to the surrounding prostate; however, statistically, there was no significant difference in the T2w SIs of the CG between the baseline and any time point after embolization (*p* = 1.00).

### 3.4. Clinical Indexes

The mean pretreatment IPSS was 21.7 ± 2.0, which decreased on average by 32.5% at one month, by 11.5% at six months, and by 15.9% at 12 months (Table 6).

The results of multiple comparisons revealed significant differences between the IPSS at the baseline and one, six, or 12 months after embolization for the total cohort and for the subgroup with baseline TGV ≤ 60 mL (Table 6), but not for the cohort of patients with baseline TGV > 60 mL. The ICIQ-UI SF score significantly decreased from 6.4 ± 1.2 at the baseline to post-treatment values of 2.4 ± 0.9 and 2.3 ± 0.8 (*p* = 0.016) at one and six months, respectively, for the total cohort. In both subgroups with baseline TGV ≤ 60 mL and >60 mL, there were similar significant differences at these time points (see Table 6). There was no significant difference of the ICIQ-UI SF score between the baseline and 12 months after PAE for the total cohort and the subgroup with baseline prostate volume > 60 mL (*p* > 0.05; Table 6).

The correlation analysis for the total cohort found that changes in TGV between the baseline and one month post-treatment correlated moderately with IPSS and ICIQ-UI SF score reduction (*p* = 0.031 and *p* = 0.05, respectively; correlation coefficients were 0.52 and 0.69, respectively). The higher the volume reduction of TG, the greater decrease in both questionnaire scores was observed.

## 4. Discussion

Several studies demonstrated that PAE is able to deliver excellent clinical outcomes in terms of prostate volume reduction, Qmax, PVR, QoL, as well as various standardized questionnaires, such as IPSS, ICIQ-UI, and IIEF [12,20,21]. Regarding changes in prostate MRI over time after PAE, only manual assessments have been performed to date, both with respect to volumetric characteristics and morphologic changes in SIs. The volume analyses of the prostate were regularly conducted with the ellipse formula by Sosna et al. [19], while, for the morphological changes, SIs were routinely measured by means of manually defined regions of interest (ROIs) placed in the infarct areas formed after PAE [17,18]. Thus, we aimed to investigate the semi-automated approach as an alternative method to assess prostate MRI after PAE, including the identification of potential benefits and the correlation with clinical parameters.

In this study, PAE was successfully performed in a cohort of 27 patients. with significant improvement in IPSS and ICIQ-UI SF one and six months after embolization compared with baseline values.

All patients presented with infarct areas occurring exclusively in the CG, which may be predominantly explained by the prostatic artery vascularization. As several studies have shown, the prostatic artery divides into a lateral branch and a medial branch [22,23]. While the lateral branch runs laterally to the apex of the prostate gland, providing branches that perforate the organ and extend into the peripheral gland, the medial branch can be found between the base of the bladder and the gland, supplying blood to most of the CG via small perforating arteries [24]. In our opinion, the infarctions only seen in the CG may be due to the fact that there is no collateral supply, while there remains a certain collateral supply in the periphery, which prevents complete necrosis there. We assessed the volumetric changes in TG and CG one, six, and 12 months following PAE and found that the TGV and CGV reduction was most obvious one month after embolization, irrespective of the baseline TGV (≤ 60 mL or >60 mL). Our findings are consistent with other studies that report the most significant volume reduction in MRI examinations one month after PAE [12,17,25]. These volumetric changes result from ischemic necrosis due to arterial stasis after the injection of embolization particles [26]. It has previously been described that prostate volume may increase over time after the PAE and clinical symptomatology may recur, similarly to our results [27]. We found that volumetric changes between the baseline and one month post-treatment correlated moderately with both clinical questionnaire scores, highlighting this timely clinical response. Partial incontinence symptoms returned at the 12-month follow-up in our cohort; nevertheless, at this time point, the IPSS remained within the definition of meaningful changes according to Barry et al. and Blanker et al. (a 3-point or 5-point reduction, respectively) [28,29]. Moreover, there was no significant difference between the semi-automatic assessed TGV and the conventional TGV calculated via the ellipse formula; thus, both volumetric measurement methods may be appropriate for an objective evaluation of the treatment success. However, further studies with larger patient groups should be performed in order to verify the equivalence of the two methods.

IPP in BPE occurs as the prostate enlarges into the bladder along the plane of least resistance. Thus, IPP indicates the presence of a median lobe enlargement without relevant involvement of the lateral lobe, which can cause bladder neck protrusion in relatively small prostate glands and has been associated with BOO [18,30,31]. This correlation was first described by Chia et al. in 2003 based on ultrasound measurement of IPP [32]. Several studies regarding IPP presented a positive correlation with prostate volume, bladder compliance, detrusor overactivity, detrusor pressure at maximum urinary flow, BOO index, and PVR, as well as a negative association with Qmax [33,34]. Lim et al. reported a significant reduction in the baseline IPP (16 mm) three months after PAE in 18 patients (13 mm) [35]. In our cohort, even if we focused on patients with small- and medium-volume BPH, 93% of patients demonstrated IPP at the baseline; in addition, a significant IPP reduction was observed for the total cohort and in the subgroup with baseline TGV > 60 mL, the latter presenting a coronary baseline IPP of 22 mm. The differences were not significant in the subgroup with baseline TGV ≤ 60 mL, explained by the less prominent baseline IPP of the smaller prostates (14 mm) and thus a decreased percentage IPP reduction.

In addition to volumetric parameters, several other MRI features following embolization have already been reported [17,36,37]. Ali et al. showed that 79% of patients demonstrated decreased T2w SIs after PAE, and 51% showed a decrease in enhancement (48%) at the six-month follow-up. A signal loss on MRI is generally a sign of degeneration or necrosis [18,38]. Zhang et al. focused on the assessment of SI changes in PAE-induced infarcts and their characteristics on T2w, T1w, and ADC-Map images. They described statistically significant differences in ADC values (b = 1000/2000/3000 s/mm^2^) after one, three, six, or 12 months. In our study, we used the most common b value in clinical routine (b = 1000 s/mm^2^) as the latter is in accordance with the current recommendations for prostate multiparametric MRI protocol by the European Society of Urogenital Radiology [39,40]. We observed a significant decrease in ADC values of the semi-automatic segmented CG before and one month after embolization. The different SIs on DWI can reflect focal changes at the cellular, perfusion, and molecular level suggestive of prostatic infarctions [41]; as DWI principally measures Brownian motion, referred to as diffusibility, the decrease in ADC values in the early stage after PAE can be explained due to postprocedural hemorrhage and edema limiting the movement of water molecules [42,43]. Until now, the infarct areas after PAE have always been identified separately and defined ROIs have been placed in them; however, we found that changes in SIs can also be detected using a semi-automatic approach while assessing the entire CG. Further studies with regard to a potential clinical benefit as a predictor for treatment response or recurrence rate are also needed. It would be interesting to see if there is a correlation between the ADC values of the CG prior to embolization and the IPSS reduction after PAE or between the differences in ADC values one month after PAE and recurrence rates later on. There are no significant changes in ADC values in TG before and after treatment based on the characteristics of blood supply and microcirculation in the peripheral zone, and prostatic infarcts only occurred after PAE in the CG [17,44]. Lastly, we found no significant differences in SIs on T1w or T2w images between the baseline and after treatment. This may also be because of the assessment itself as the sensitivity of the approach may not be sufficient while measuring the entire CG in the form of semi-automatic segmentation and not the SIs of the manually separated infarct zones.

The present study has several limitations. One of the main limitations is the small sample size. In addition, the study was monocentric, and PAE was routinely performed with a certain technical approach, including the use of one distinct particle type for embolization. Though this may not significantly change the core findings of this study, further studies with a higher number of patients, involving several centers and/or different technical approaches with various embolic agents, are required to validate the results. Lastly, the follow-up period was limited to 12 months; thus, particularly regarding recurrences, which potentially occur in MRI and clinically after a longer time, this is a rather short period that risks misrepresenting the treatment success. Nevertheless, the data evaluated in this study at the follow-up were acquired prospectively and at defined time points.

## 5. Conclusions

In conclusion, the semi-automatic approach is an adequate method for assessing volumetric and morphological changes in prostate MRI following PAE because significant differences in ADC values can be measured and there is no need for manual identification of infarct areas. The volumetric assessment using the semi-automatic approach revealed significant reductions in CGV and TGV, the latter comparable to calculation by the conventional ellipse formula. This volume reduction confirmed the clinical response after PAE highlighted by the correlated changes in the clinical scores (IPSS, ICIQ-UI SF). The minimal increase in prostatic volume and IPP, as well as the recurrence of clinical symptoms at the 12-months follow-up, indicate slightly decreased long-term success compared to the short-term results.

## Figures and Tables

**Figure 1 diagnostics-12-00585-f001:**
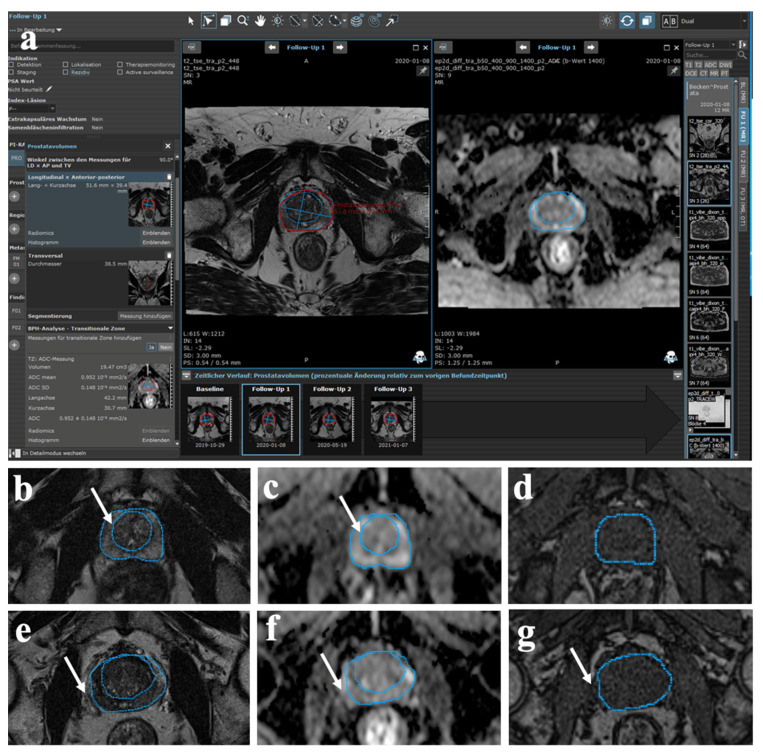
Methodology. MRI analysis using the dedicated semi-automatic software tool mint Lesion™. (**a**) Software surface for the systematic and standardized assessment of volumetric and morphological characteristics after PAE. (**b**,**e**) Axial T2w MR images after segmentation of the CG (arrow in (**b**)) and the TG (arrow in (**e**)) performed for each slice, respectively. (**c**,**f**) Subsequent automatic transfer of both measurements to the corresponding slice levels of the axial DWI sequence (arrows). (**d**,**g**) Subsequent automatic transfer of the measured TG to the corresponding slice levels of the axial T1w MRI sequence (arrow).

**Figure 2 diagnostics-12-00585-f002:**
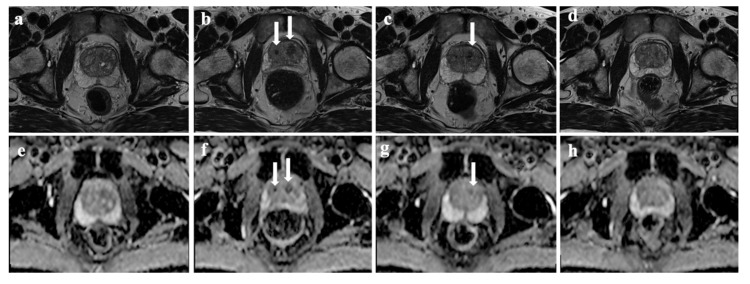
CG infarction. Prostate MRI of a 75-year-old patient with BPH. Axial T2w images and axial DWI of the prostate at baseline (**a**,**e**) and one, six, and 12 months (**b**–**d**,**f**–**h**) after PAE show bilateral infarct areas, which are most clearly delineated one month after embolization (arrows) and are increasingly minor manifested after six and 12 months (arrow).

**Table 1 diagnostics-12-00585-t001:** MRI acquisition parameters.

Protocol	Sequence	TR (ms)	TE (ms)	FA	FOV (cm)	Matrix	Slice Thickness (mm)	NEX	Bandwidth (Hz/Pixel)
Axial T2w	Turbo SE	8900	119	159	20 × 24	448 × 340	3	2	200
Sagittal T2w	Turbo SE	12,500	114	144	20 × 20	320 × 320	3	2	200
Axial T1w opposed phase	VIBE	12.5	1.3	9	33 × 40	320 × 221	2	1	1040
Axial T1w in phase	VIBE	12.5	2.5	9	33 × 40	320 × 221	2	1	1040
Axial T1w fat	VIBE	12.5	1.26	9	33 × 40	320 × 221	2	1	1040
Axial T1w water	VIBE	12.5	1.26	9	33 × 40	320 × 221	2	1	1040
Axial DWI trace-w (b = 50 s/mm^2^)	EPI SE	47,917	61	90	19 × 24	96 × 57	3	16	1795
Axial DWI ADC (b = 1000 s/mm^2^)	EPI SE	47,917	61	90	19 × 24	96 × 57	3	16	1795
Axial DWI (b = 1600 s/mm^2^)	EPI SE	47,917	61	90	19 × 24	96 × 57	3	16	1795

ADC = apparent diffusion coefficient, EPI = echo planar imaging, SE = spin-echo, VIBE = volumetric interpolated breath-hold examination, DWI = diffusion weighted imaging, TE = echo time, TR = repetition time, FA = flip angle, FOV = field of view, NEX = number of excitations.

**Table 2 diagnostics-12-00585-t002:** Baseline characteristics of study cohort.

Characteristics		Total Cohort (*n* = 27)
Age (years)	Mean (range)	67 (43–84)
PSA (ng/mL)	Mean (range)	4.6 (0.4–26.1)
Prostate arteries embolized		
Unilateral		2 (17.4%)
Bilateral		25 (92.6%)
Baseline TGV (mL) Baseline CGV (mL) Baseline IPSS Baseline ICIQ-UI SF score	Median (range) Median (range) Median (range) Median (range)	81 (31–163) 59 (15–154) 23 (5–33) 5 (0–14)

PSA = prostate-specific antigen, TGV = total gland volume, CGV = central gland volume, IPSS = international prostate symptom score, ICIQ-UI SF = international consultation on incontinence questionnaire–urinary incontinence short form.

**Table 3 diagnostics-12-00585-t003:** Pre- and post-treatment TGV using ellipse formula and semi-automatic segmentation method. Values presented as means ± SD. *P*-values include Bonferroni multiple testing correction.

Cohort	BL TGV by Ellipse Formula (mL)	Post- Treatment	TGV by Ellipse Formula (mL)	*p*-Value	BL TGV by Semi-Automatic Segmentation (mL)	Post- Treatment	TGV by Semi- Automatic Segmentation (mL)	*p*-Value
Total cohort (*n* = 27)	87.2 ± 10.2	1 month	73.6 ± 8.8	0.005	80.4 ± 10.1	1 month	69.9 ± 8.5	0.025
		6 months	71.2 ± 9.0	0.005		6 months	69.6 ± 8.9	0.025
		12 months	73.1 ± 8.9	0.005		12 months	70.2 ± 8.8	0.050
Baseline TV ≤ 60 mL (*n* = 12)	42.1 ± 4.0	1 month 6 months	35.2 ± 2.9 35.1 ± 2.9	0.050 0.050	35.6 ± 3.6	1 month 6 months	32.9 ± 3.4 31.3 ± 2.7	0.001 0.001
		12 months	35.1 ± 2.8	0.050		12 months	31.8 ± 2.5	0.001
Baseline TV > 60 mL (*n* = 15)	115.8 ± 8.5	1 month 6 months	97.7 ± 8.0 94.2 ± 9.3	0.024 0.024	110.2 ± 9.4	1 month 6 months	94.6 ± 8.1 94.2 ± 8.7	0.050 0.050
		12 months	97.2 ± 8.2	0.024		12 months	94.8 ± 8.4	0.050

**Table 4 diagnostics-12-00585-t004:** Pre- and post-treatment CGV and IPP. Values presented as means ± SD. *p* values include Bonferroni multiple testing correction.

Cohort	BL CGV Semi- Automatic Segmentation (mL)	Post- Treatment	CGV Semi-Automatic Segmentation (mL)	*p*-Value	BL IPP Coronary (mm)	Post- Treatment	IPP Coronary (mm)	*p*-Value
Total cohort (*n* = 27)	58.1 ± 9.2	1 month	47.6 ± 7.3	0.038	19.4 ± 2.3	1 month	17.1 ± 2.1	0.028
		6 months	47.8 ± 7.5	0.191		6 months	16.9 ± 1.9	0.010
		12 months	48.4 ± 7.3	0.283		12 months	17.0 ± 2.1	0.018
Baseline TV ≤ 60 mL (*n* = 12)	19.6 ± 3.2	1 month 6 months	16.7 ± 2.4 16.8 ± 2.7	0.043 0.043	14.4 ± 3.2	1 month 6 months	12.1 ± 2.3 12.1 ± 2.1	0.210 0.190
		12 months	17.3 ± 2.4	0.446		12 months	13.0 ± 2.8	1.000
Baseline TV > 60 mL (*n* = 15)	83.7 ± 9.2	1 month 6 months	68.3 ± 7.2 68.6 ± 7.7	0.050 0.346	22.6 ± 2.9	1 month 6 months	19.8 ± 2.7 19.7 ± 2.7	0.029 0.005
		12 months	69.2 ± 7.2	0.439		12 months	20.1 ± 2.8	0.154

**Table 5 diagnostics-12-00585-t005:** Pre- and post-treatment ADC values presented as means ± SD; *p*-values include Bonferroni multiple testing correction.

Cohort	Baseline ADC Value of CG (10^−3^ mm^2^/s)	Post- Treatment	ADC Value of CG (10^−3^ mm^2^/s)	*p*-Value	Baseline ADC Value of TG (10^−3^ mm^2^/s)	Post- Treatment	ADC Value of TG (10^−3^ mm^2^/s)	*p*-Value
Total cohort (*n* = 27)	1.20 ± 0.26	1 month	1.13 ± 0.23	<0.001	1.21 ± 0.32	1 month	1.18 ± 0.23	1.000
		6 months	1.19 ± 0.27	0.197		6 months	1.21 ± 0.23	1.000
		12 months	1.19 ± 0.22	0.353		12 months	1.22 ± 0.18	1.000
Baseline TV ≤ 60 mL (*n* = 12)	1.16 ± 0.39	1 month 6 months	1.08 ± 0.29 1.13 ± 0.31	<0.001 0.521	1.20 ± 0.35	1 month 6 months	1.13 ± 0.31 1.16 ± 0.19	0.529 1.000
		12 months	1.14 ± 0.34	0.541		12 months	1.18 ± 0.31	1.000
Baseline TV > 60 mL (*n* = 15)	1.25 ± 0.29	1 month 6 months	1.17 ± 0.30 1.23 ± 0.35	<0.001 0.280	1.21 ± 0.47	1 month 6 months	1.20 ± 0.29 1.24 ± 0.32	1.000 1.000
		12 months	1.23 ± 0.22	0.517		12 months	1.24 ± 0.21	1.000

**Table 6 diagnostics-12-00585-t006:** Pre- and post-treatment IPSS and ICIQ-UI SF scores. Values presented as means ± SD; *p*-values include Bonferroni multiple testing correction.

Cohort	BL IPSS	Post- Treatment	IPSS	*p*-Value	BL ICIQ-UI SF Score	Post- Treatment	ICIQ-UI SF Score	*p*-Value
Total cohort (*n* = 27)	21.8 ± 2.0	1 month	14.7 ± 2.1	0.002	6.4 ± 1.2	1 month	2.4 ± 0.9	0.016
		6 months	11.5 ± 1.8	<0.001		6 months	2.3 ± 0.8	0.020
		12 months	15.9 ± 1.9	0.003		12 months	3.5 ± 1.0	0.061
Baseline TV ≤ 60 mL (*n* = 12)	23.3 ± 2.8	1 month 6 months	17.8 ± 3.1 14.5 ± 2.9	0.050 0.018	6.0 ± 2.2	1 month 6 months	2.0 ± 1.4 1.6 ± 0.9	0.012 0.007
		12 months	17.1 ± 3.0	0.049		12 months	1.7 ± 1.1	0.009
Baseline TV > 60 mL (*n* = 15)	20.6 ± 2.9	1 month 6 months	12.2 ± 2.8 9.1 ± 2.1	0.022 0.016	6.7 ± 1.5	1 month 6 months	2.8 ± 1.3 2.9 ± 1.2	0.038 0.042
		12 months	15.0 ± 2.4	0.180		12 months	4.8 ± 1.5	0.073

## Data Availability

Data are available on request.

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
