# Peer review of "Semi-Automatic MRI Feature Assessment in Small- and Medium-Volume Benign Prostatic Hyperplasia after Prostatic Artery Embolization"

_diagnostics, 2022, doi:10.3390/diagnostics12030585_

Round 1
Reviewer 1 Report
The term Benigne Prostate Hyperplasia (BPH) identifies a condition of histological epithelial and stromal cells proliferation which determinates a progressive enlargement of the prostate, especially of the Transition Zone (ZT). This pathological status leads to an increasingly obstruction of the urinary flow and it is linked with several bladder voiding and/or filling phase symptoms.
There are different approaches to this condition and Prostatic Artery Embolization (PAE) seems to represent a very valid one.
The aim of this study is to find and validate an alternative semi-automatic method to assess prostate MRI after PAE, including the identification of potential benefits and the correlation with clinical parameters. To do that, authors enrolled in this subgroup analysis within the ongoing prospective clinical trial PROEMBO a total of 27 patients with severe BPH-related LUTS and IPSS ≥8.
The authors should be congratulated for the great work and the interesting topic discussed. The manuscript is well-written, easily readable, but it is lacking in some points that would add value to the entire manuscript:
Introduction. Referred to lines 57-58: it is worth mentioning which articles support these data. This article (https://doi.org/10.1159/000516681) could strengthen the value of the sentence and of the entire manuscript. The correlation rate between MRI and a specific prostatic condition is offered by the results of this study too (https://doi.org/10.1007/s00261-020-02798-8). Authors could benefit from a reading of them.
Methods. Referred to lines 99-100: authors should elucidate why they choose IPSS and ICIQ-UI SF as clinical indexes among all.
The methodology was robust.
Discussion. Authors should discuss whether they believe that overcoming such limitations, such as the number of patients, may have some effect on the results.
A revision of the language is recommended.
Reviewer 2 Report
Please attached file. I have gone through line by line and provided overall comments and questions, with major and minor concerns.

Reviewer 3 Report
The article is well constructed. The benefit for the collectivity is not great to date.
The main goal of PAE is clinical improvment. Reduction of IPSS is always correlate with MRI observations. A very interesting analysis would have been the gland signal analysis prior to the PAE in order to find predictive parameters of good response to the embolization. The lower the pre embolization ADC is, the worst is the IPSS reduction ? Semi automatic analysis could help to find these kind of informations.
Re arrange table 1 in order to make it readable, columns are not aligned.
In the table 2 : some patients seem to be no includable because of the low IPSS (5). It probably correspond to the low PSA rate of 0.5 ng/ml.
You should explain why some patients like this have finally been included.
For small prostate, the assesment of volume reduction after embolization could be challenging, therefore a semi automatic analyse can improve confidence.
Round 2
Reviewer 1 Report
Authors correctly answered comments.